# Development and Themes of Diagnostic and Treatment Procedures for Secondary Leg Lymphedema in Patients with Gynecologic Cancers

**DOI:** 10.3390/healthcare7030101

**Published:** 2019-08-27

**Authors:** Yumiko Watanabe, Masafumi Koshiyama, Keiko Seki, Miwa Nakagawa, Eri Ikuta, Makiko Oowaki, Shin-ichi Sakamoto

**Affiliations:** 1Department of Women’s Health, Graduate School of Human Nursing, The University of Shiga Prefecture, Shiga 522-8533, Japan; 2School of Human Nursing, The University of Shiga Prefecture, Shiga 522-8533, Japan; 3School of Engineering, Department of Electronic Systems Engineering, The University of Shiga Prefecture, Shiga 522-8533, Japan

**Keywords:** diagnosis, treatment, lymphedema, gynecologic cancer

## Abstract

Patients with leg lymphedema sometimes suffer under constraint feeling leg heaviness and pain, requiring lifelong treatment and psychosocial support after surgeries or radiation therapies for gynecologic cancers. We herein review the current issues (a review of the relevant literature) associated with recently developed diagnostic procedures and treatments for secondary leg lymphedema, and discuss how to better manage leg lymphedema. Among the currently available diagnostic tools, indocyanine green lymphography (ICG-LG) can detect dermal lymph backflow in asymptomatic legs at stage 0. Therefore, ICG-LG is considered the most sensitive and useful tool. At symptomatic stage ≥1, ultrasonography, magnetic resonance imaging-lymphography/computed tomography-lymphography (MRI-LG/CT-LG) and lymphosintiography are also useful. For the treatment of lymphedema, complex decongestive physiotherapy (CDP) including manual lymphatic drainage (MLD), compression therapy, exercise and skin care, is generally performed. In recent years, CDP has often required effective multi-layer lymph edema bandaging (MLLB) or advanced pneumatic compression devices (APCDs). If CDP is not effective, microsurgical procedures can be performed. At stage 1–2, when lymphaticovenous anastomosis (LVA) is performed, lymphaticovenous side-to-side anastomosis (LVSEA) is principally recommended. At stage 2–3, vascularized lymph node transfer (VLNT) is useful. These ingenious procedures can help maintain the patient’s quality of life (QOL) but unfortunately cannot cure lymphedema. The most important concern is the prevention of secondary lymphedema, which is achieved through approaches such as skin care, weight control, gentle limb exercises, avoiding sun and heat, and elevation of the affected leg.

## 1. Introduction

After radical surgeries for cervical, endometrial and ovarian cancers, patients often suffer from chronic leg lymphedema. These patients therefore often suffer leg heaviness, erythema, ulcers and pain, requiring lifelong treatment and psychosocial support [1]. Such lymphedema is the result of lymphatic system insufficiency and impaired lymph transport due to lymphadenectomy or radiation therapy [2].

Ki et al. reported that the incidence of lower-limb lymphedema (LLL) was 11.1% after surgery for ovarian cancer [3]. In Japan, 16.6% of patients with cervical cancer suffer from LLL after pelvic lymph node removal [4]. In a prospective longitudinal cohort study (*n* = 408), Hayes et al. reported that 50% of patients showed evidence of lymphedema within 2 years post-gynecological cancer, and 60% of lymphedema cases were persistent [5]. LLL is a common long-term complication that reduces the quality of life (QOL) of gynecological cancer survivors [6,7]. Evidence that the disease is often overlooked by physicians caring for such patients lies in the fact that about 60% of patients are self-referred for an initial evaluation after several years [8,9]. Therefore, medical teams must address lymphedema as a part of the early treatment of gynecologic cancers.

We herein review the current issues (with a review of the relevant literature) and etiology, symptoms, recent diagnostic procedures and treatments for secondary leg lymphedema in patients with gynecologic cancers and discuss how to better manage leg lymphedema after surgery or radiation therapy.

## 2. Etiology and Pathophysiology of Leg Lymphedema after Pelvic Lymphadenectomy

Secondary lymphedema is a chronic disease characterized by the accumulation of interstitial fluid in tissues due to damaged lymphatic vessels, leading to leg swelling and dysfunction [10]. The risk of leg lymphedema has been correlated with the number of lymph nodes removed and the location of the removed lymph nodes. For example, Todo et al. revealed that removal of the circumflex iliac nodes to the distal external iliac nodes (CINDEINs) can be a critical risk factor for postoperative lower-extremity lymphedema [11]. They also revealed that adjuvant whole pelvic external beam radiation therapy, resection of ≥31 lymph nodes and removal of the CIDENs were significantly related to its occurrence. Similarly, Togami et al. found that the number of lymph nodes removed (odds ration [OR]: 3.37, 95% confidence interval [CI]: 1.43–8.54; *p* = 0.0053) and circumflex iliac node (CIN) removal (OR: 3.92; 95% CI: 1.55–11.4; *p* = 0.0033) were independent risk factors for lower extremity lymphedema [4]. In contrast, Hopp et al. suggested that the development of lymphedema may be a process not influenced by the number or location of lymph nodes removed, surgical approach, or medical comorbidities [12].

Under normal conditions (before lymphadenectomy), reabsorption of extracellular fluid and protein occurs via both the arteriovenous and the lymphatic capillaries [13]. An extracellular flow pattern occurs along the connective tissue fiber, and presumably, through prelymphatic channels [14]. Lymphatic channels primarily regulate the flow of interstitial fluid [15]. Venous capillaries reabsorb 90% of the interstitial fluid and lymphatic channels absorb the remaining 10% of lymph fluid and proteins [16]. Fluid and particles enter the initial lymphatics through interendothelial openings and by vesicular transport through the lymphatic endothelial cells [17]. Intracellular transport occurs through the phagocytosis of macrophages entering the lymphatics.

The main propelling forces for lymph flow are the rhythmic contractions of lymphangions (segments of lymphatics between two unidirectional valves), which generate lymph pressures high enough to move the intralymphatic fluid centripetally [18]. Muscular contractions, respiratory movements and arterial pulsations play a secondary role to the spontaneous contractions of lymphatics [19]. Lymphatic contraction can be regulated by the nervous system, as both noradrenergic and sensory nerve endings have been found in the lymphatic wall [13]. Noradrenaline, serotonin, prostaglandin F2α, thromboxane B2 and endothelin 1 all contract lymph vessels [20,21]. Lymphatic fluid then passes to the regional lymph nodes and is transported back into the subclavian vein to enter the venous system via the thoracic duct [22].

Pelvic lymphadenectomy or irradiation can induce the destruction or obstruction of the central lymphatic vessels. Leg lymphedema is thus caused by the excess accumulation of tissue fluid and lymph resulting from a lack of proper centripetal drainage due to damage to the lymphatic collector wall [18]. In other words, endolymphatic pressure increases in the lymphatic vessels of the leg due to the lack of proper centripetal drainage and histological changes in the vessels ensure which causes protein-rich fluid to accumulate in the interstitial spaces of the leg [23]. Lymphatic capillary hypertension in primary lymphedema has been demonstrated [24,25]. Olszewski measured the end pressures of the lymphatic ducts in a lymphedematous human leg during ambulation (up to 200 mmHg) [26]. A reduced flow or stasis of lymph can result from lymphatic hypoplasia, from the obliteration of lymphatic trunks, from the absence of lymphatic valves, or from impaired lymphatic contractility [27,28]. Lymph stasis results in the accumulation of protein in the extracellular space, which increase the tissue colloid osmotic pressure, causing edema formation and the elevation of interstitial hydraulic pressure (Figure 1) [29,30]. The potential role of interstitial glycosaminoglycans within the skin and subcutaneous tissue in the pathogenesis of lymphedema has been a point of focus [31].

Over time, the inflammatory lymphatic fluid starts to damage the natural lymphatic drainage pathways and surrounding tissue [32]. The accumulation of proteins attracts macrophages, stimulates collagen production by fibroblasts and enhances the stimulation of fibroblasts, keratinocytes and adipocytes [28]. In brief, leg lymphedema is a chronic, debilitating condition characterized by the abnormal accumulation of proteinaceous fluid in skin and subcutaneous tissue resulting in subsequent adipose and fibrous tissue hyperplasia [33,34].

In secondary lymphedema, the immune function is significantly compromised. The interaction of CD4^+^ T cells and macrophages has been shown to play a role in driving the proliferation of lymphatic endothelial cells and aberrant lymphangiogenesis, which contribute to the interstitial fluid accumulation in lymphedema [35]. CD4^+^ T cells are activated in skin-draining lymph nodes and then migrate to lymphedematous skin [36]. Activated CD4^+^ T cells promote fibrosis and inflammation, and inhibit lymphangiogenesis and the lymphatic function.

## 3. Symptoms

The clinical severity of the disease symptoms is classified according to the International Society of Lymphology (ISL) and divided into stages 0 to 3 (Table 1) [37]. A latent or subclinical condition is defined as stage 0. The early accumulation of fluid that subsides with limb elevation is defined as stage 1. The transition from stage 0 lymphedema to stage 1 is defined as the point at which a detectable change in the limb volume occurs [38]. The condition in which limb elevation alone rarely reduces the tissue swelling and pitting is defined as stage 2. The condition in which the limb may not show any pitting, as excess subcutaneous fat and fibrosis develop, is defined as later stage 2. As the physical properties of the interstitium change during the course of the disease, these changes will alter the hydraulic conductivity of the tissue and thus, both the clearance of fluid and the lymphangiogenic response [39,40]. The mechanical changes in the tissue itself (i.e., fibrosis) are a hallmark of disease severity and disease progression to an irreversible state at stage 2 [41,42]. The condition in which elephantiasis develops due to alterations in the skin character and thickness is defined as stage 3. Given that inflammatory cytokines have also been shown to affect the collecting lymphatic pump function [43,44], it is likely that the consequences of inflammatory-driven lymphedema progression not only alter the interstitium, but severely compromise the pump function and drainage by the collectors at stages 2 and 3 [38].

Elephantiasis occurs at stage 3. The main symptom of elephantiasis is gross enlargement and swelling of the legs because of the accumulation of fluid. The entire leg may swell to several times its normal size resembling the thick, round appearance of an elephant’s leg. The skin of the leg usually develops a dry, thickened and pebbly appearance. It presents as diffuse non-pitting edema, ulceration and a hyperkeratotic papulonodule with a verrucous appearance on the leg [45].

Cellulitis with lymphangitis sometimes occurs when bacteria or toxin absorbed into wounds or skin infections spread into the subcutaneous tissue of the leg lymphoma [46]. For example, a 47.6% prevalence rate of cellulitis has been observed among patients in Thailand with lymphedema [47]. Cellulitis starts with by flu-like symptoms followed by leg erythema and swelling that presents as a hot sensation or tenderness [48]. Cellulitis is mainly caused β-hemolytic streptococci invasion into the lower extremities [46].

## 4. The Diagnosis

Diagnostic procedures for secondary leg lymphoma and their features are summarized in Table 2.

### 4.1. Physical Examination Findings

Characteristic skin changes, such as edema, peau d’orange, cutaneous fibrosis, and a positive “Stemmer sign”, help diagnose lymphedema [9]. The positive Stemmer sign presents as the inability to pinch a fold of skin at the base of the second toe due to swelling in the subcutaneous layer and thickening of the skin [49].

In the early stages of lymphedema, pitting edema also occurs, as with common edema, because of an influx of protein-rich fluid into the interstitium before fibrosis of the subcutaneous tissue occurs, and the skin has a doughy appearance; in contrast, in the later stages, it becomes fibrotic, thickened and verrucous [50].

### 4.2. Ultrasonography

We recently showed that portable B-scan ultrasonography was able to quantitatively measure the increased thickness of the skin of the leg in pregnant women [51]. The skin thickness of the legs in pregnant women with edema was significantly increased compared with that in pregnant women without edema (6.4 ± 0.3 mm vs. 4.6 ± 0.4 mm) (*n* = 98; *p* = 0.0001). Using ultrasonography, Suehiro et al. also examined the skin and subcutaneous tissue in secondary LLL [52]. They demonstrated three typical ultrasound findings in extremities with lymphedema namely, increased skin thickness, increased subcutaneous tissue thickness, and increased subcutaneous echogenicity. All three findings correlated well with the ISL stage. They additionally described additional findings of ultrasonography in legs with secondary lymphedema in the next year [53]. They also showed that echogenicity was increased in the medial thigh and legs, while remarkable echo-free space (EFS) was observed only in the lower leg. As the clinical severity progressed, the echogenicity increased in all parts of the lower extremity. EFS also increased in all parts of the leg, but the lower leg still showed the most severe condition. As above, we believe that ultrasonography may be a convenient tool for assessing the degree of lymphedema.

### 4.3. Magnetic Resonance Imaging (MRI)

Hoffner et al. examined the limbs in seven patients with arm lymphedema and six with leg lymphedema using magnetic resonance imaging (MRI) [54]. They examined the limbs with water-fat MRI before and after liposuction. Water-fat MRI failed to differentiate water signals between fluid and muscle, which made it difficult to evaluate subfacial tissue changes. This modality therefore cannot detect superficial tissue changes, or changes in the whole composition of the extremity. MRI therefore is not suitable for the diagnosis of lymphedema changes.

### 4.4. MRI-Lymphography/Computed Tomography-Lymphography (MRI-LG/CT-LG)

Indirect MRI-LG was technically feasible, and no complications were observed after the intracutaneous injection of gadodiamide [55]. Relatively high-definition images can be obtained without radiation exposure using MRI-LG [56]. However, there are some limitations associated with MRI-LG, such as the long duration (about an hour) of the MR examination and occasional difficulty in distinguishing the affected lymphatic vessels when an underlying remarkable venous contamination is present [57]. In addition, the contrast agent used for MRI can easily enter the venous circulation [58]. Indeed, White et al. reported the need for an intradermal injection rather than a subcutaneous injection for the optimal visualization of lymphatics and poor venous contamination [59].

Suga et al. first reported CT-LG using animals in 2003 [60]. They reported that CT-LG using small volumes of iopamidol can sufficiently visualize lymphatic drainage and may have potential utility for sentinel lymph node mapping. CT-LG images can be obtained within a short time (about 5 min), and the imaging devices are relatively widely distributed [56]. However, CT-LG has a disadvantage in its associated radiation exposure. Yamada et al. reported that CT-LG had better resolution than lymphoschintiography and enabled the clear visualization of lymphatic vessels with a minimum lumen size of 0.7 mm [56]. CT-LG also shows the three-dimensional architecture of the dermal back-flow of lymph, which originates from deep lymphatic collectors via branched small lymphatic vessels.

### 4.5. Lymphosintiography

Lymphoscintigraphy, the radionuclide technique of imaging the lymphatic system using interstitially injected radiopharmaceutical particles, was first introduced in 1953 [61]. This minimally invasive procedure simply requires intradermal or subcutaneous injection of the chosen radiolabeled tracer [62].

The protocol for lymphoscintigraphy is not standardized and differs in the choice of radiotracer, the type and site of injection, the use of dynamic and static acquisitions, and the acquisition times.

Regarding the best injection technique for finding lymphatic leakage, the subcutaneous route is better than the intradermal one due to its dual uptake mechanisms [62]. However, both subcutaneous and intradermal injections are used in studies of superficial lymphatics of the extremity. ^99m^Tc dextran is the radiotracer usually used for lymphoscintigraphy.

On lymphoscintigrams of lymphedema with abnormal findings, a variety of findings can be identified, including interruption of lymphatic flow, collateral lymph vessels, dermal backflow, delayed flow, delayed visualization or nonvisualization of lymph nodes, a reduced number of lymph nodes, dilated lymphatics and in severe cases, no visualization of the lymphatic system [63,64,65]. Lymphoscintigrams of patients with primary lymphedema reportedly tend to show a lack of lymphatic vessels and absent or delayed transport, whereas those of patients with secondary lymphedema tend to show obstruction with visualization of discrete lymphatic trunks and slow transport [65,66].

### 4.6. Indocyanine Green Lymphography (ICG-LG)

The early diagnosis of secondary lymphedema by inspection and palpation is very difficult. Lymphoscintigraphy has therefore been established as the main approach for the diagnosis of lymphedema [61,62,63,64,65,66]. However, the definite diagnosis of early-stage lymphedema is also difficult because edema progresses very slowly [67]. The development of secondary lymphedema starts several years after cancer therapy and shows within-day variation in the early stage.

The newer technique of fluorescence lymphography using ICG has led to an improvement in the diagnosis of lymphedema [67,68]. ICG-LG clearly visualizes the superficial lymph flow in real time without radiation exposure, and recently has been used in the evaluation of lymphedema [69]. Yamamoto et al. investigated the lymphatic architectural changes in asymptomatic lower extremities of patients after gynecological and urological cancer surgeries, using ICG-LG [68]. They detected the dermal lymph backflow sign in asymptomatic limbs and found that the splash pattern is the earliest finding on ICG-LG of asymptomatic limbs (at stage 0) of secondary lower extremity lymphedema patients. They conducted the leg dermal lymph backflow stage (Table 3 [70]), which can allow early diagnosis of secondary lower extremity lymphedema even in a subclinical stage [70]. Suami et al. further suggested that ICG-LG allows for the prompt identification of lymphatic vessels and has the potential to improve the outcomes of lymphovenous shunt operations and be useful as a diagnostic tool [71].

## 5. Treatments

The current standard treatments for lymphedema are complex decongestive physiotherapy (CDP) including manual lymphatic drainage (MLD), compression therapy, exercise and skin care. CDP, mainly involving MLD is separated into the intensive phase and the maintenance phase. The intensive phase of treatment comprises a course of daily exercise and MLD to decongest the lymphedematous area of the leg, followed by multiple-layer short stretch bandaging to prevent the reaccumulation of fluid and skin care [7]. The cardinal principles of treatments for lymphedema of international consensus were shown in “Best Practice for The Management of Lymphoedema” [72]. In this section, we review the currently available methods for treating (MLD, bandaging, pneumatic compression therapy, microsurgery [PCT], lipectomy and surgical excision) and preventing lymphedema.

These treatments for leg lymphedema are summarized in Table 4.

### 5.1. MLD

In the case of watery leg edema, there are two major ways of achieving interstitial fluid movement to manage subcutaneous edema of the leg and foot [73]. One way involves encouraging fluid movement in the extravascular space. Another involves encouraging fluid movement from the extravascular space into the venous system. In cases of leg lymphedema, interstitial fluid movement should be encouraged in the extra vascular and extra-lymphvascular spaces. Thus, MLD aims to encourage fluid away from congested areas by increasing the activity of normal lymphatics and bypassing ineffective or obliterated lymph vessels [72]. Deep, heavy-handed massages should be avoided in order not to damage the subcutaneous tissues and exacerbate edema by increasing capillary filtration. In accordance with “Best Practice for The Management of Lymphoedema”, MLD should be performed for up to an hour daily, with the patient lying down, in a proximal-to-distal manner in order to drain the affected areas [72].

Liao et al. reported that the key to predicting successful treatment of leg lymphedema is the initial percentage of excess volume (PEV) [7]. They found that the percentage reduction of excess volume (PREV) was not correlated with the number of CDP sessions involving MLD. This is because most of the lymphedema volume reduction occurs in the first 10 days of CDP therapy involving MLD, after which much less reduction may be achieved [74]. PEV was the only predictor of PREV, indicating that a lower PEV (early stage of lymphedema) predicts a better response to CDP therapy. Treatment for lymphedema should therefore be started at early stage to completely eliminate lymphedema. When started at an advanced stage, the swollen leg, once its subcutaneous tissues have been destroyed, cannot easily return to its former state. However, Yamamoto et al. reported that removing only the pelvic lymph nodes resulted in better CDP efficacy (good PREV) than removing both the pelvic and para-aortic lymph nodes [75].

MLD treatments should be delayed if cancer patients are receiving chemotherapy, particularly as the chemotherapy itself may alleviate obstructive edema by reducing the tumor size. However, it may not be appropriate to delay lymphedema treatment, as swelling may become poorly controlled [76]. In the presence of acute infection, MLD should not be performed. Once antibiotic therapy has begun and systemic symptoms have passed, MLD can be considered [76].

### 5.2. Bandaging

Bandaging is also an important component of CDP. Multi-layer lymph edema bandaging (MLLB) is a key element of intensive therapy regimens [72]. Inelastic bandaging is generally used, which has low extensibility and produces high working pressures and low resting pressures, thus stimulating lymph flow. However, Hara et al. indicated that the pressure of MLLB varied among therapists and training courses [77]. Furthermore, they found that the most suitable pressure for lymphedema treatment is 50–59 mmHg, which is only achieved by 27.1% of therapists. They thus insisted on the need for a uniform curriculum of training courses including measurement of the bandaging pressure. Erglin et al. researched the ideal duration for applying bandaging [78]. They divided the average number of hours of bandage compliance into 2 groups: 13–24 and 7–12 h. No significant difference in the volume reduction was noted between the two groups. This means that 12 h is enough time to bandaging, as keeping bandages on for 12–24 h has the same effect on patients as receiving CDP.

Recently, the specific effects of nighttime multilayer bandaging (NMLB) alone have been reported [79], with NMLB significantly reducing both swelling and swelling-related symptoms of cancer-related lymphedema.

Compression garments are another approach to compression therapy for lymphedema. The benefits of such garments are a comfortable fit, adequate graduated compression to manage edema, an acceptable design for the patient’s lifestyle, and manageable cost [80]. The key point of compression garments is to counteract the force of gravity and promote the normal flow of venous blood up the leg, thereby improving venous and lymph return and reducing edema.

### 5.3. PCT

The standard of care, such as CDT, includes skin care, MLD, compression bandaging, compression garments and the selected use of pneumatic compression devices (PCDs) [81,82]. PCDs for lymphedema treatment usually have multiple chambers (typically 3–10) that provide sequential treatment and may induce gradient pressure with limited adjustability [83]. In addition, advanced PCDs (APCDs) provide calibrated, gradient sequential inflation with custom treatment to address individual patients’ needs. Recently developed APCDs have a programmable controller, which offers several treatment programs to address lower extremity swelling, producing a gentle, wave-like application of pressure as the chambers sequentially inflate and deflate [83]. The sleeve embraces the whole limb up to the inguinal crease. The device has multiple-chambers, sequential inflation, gradient inflation pressure, sufficient inflation timing for the translocation of tissue fluid to the proximal region, no deflation of distal chambers in order to prevent fluid back-flow, and venous stasis in the superficial limb system [84].

Zaleska et al. showed that long-term external limb compression is followed by increase in tissue channel cross surface area in the thigh, with concurrent decrease of calf circumference [84]. They suggested that the new channels developed proximally enabling the moved fluid to be absorbed in the upper thigh, hip, and lumbar with a normal lymphatic drainage. Aldrich et al. also provided evidence that PCT may facilitate lymph vessel uptake and transport, subsequently to distal-to-proximal movement of ICG-laden lymph during a single one-hour treatment session in subjects with established lymphedema who did not previously use PCT regularly [85]. They indicated that lymphatic contractile activity in patent lymphatic vessels occurred in concert with the sequential cycling of PCT.

### 5.4. Lymphaticovenous Anastomosis (LVA) and Vascularized Lymph Node Transfer (VLNT)

LVA is a microsurgical procedure that can improve the patient’s own physiologic drainage of the lymphatic fluid. Several authors have applied LVA in several variations of end-to-end or end-to-side techniques [86,87] and described the long-term results of LVA in circumferential and volume reduction of the affected limbs [88,89]. Lymphatic and venous autografts are interpositioned to bypass lymphatic blockade [90,91]. These microsurgical procedures can be performed if some portions of the peripheral collecting lymphatics remain patent and partially drain tissue edema fluid [92]. In fact, the patients who received LVA and had records of their lower extremity circumference observed an average changing rate of −4.7% in lower extremity lymphedema index after the surgery [93]. Furthermore, LVA was able to significantly reduce pain in the limbs affected by lymphedema.

Regarding the indication of LVA, this technique may be considered for patients who are refractory to conservative treatment due to its low level of invasiveness, even if they have early-onset lymphedema. In particular, LVA is effective in cases of leg dermal backflow stage 2 and no backflow [94].

Lymphatic obstruction is more likely with lymphaticovenous end-to-end anastomosis (LVEEA) than with lymphaticovenous side-to-side anastomosis (LVSEA). Therefore, when LVA is performed, Suzuki et al. recently recommended LVSEA principally and LVEEA only when the potential for consequences and risk of obstruction are low [95]. Maegawa et al. demonstrated midterm postoperative patency of a LVSEA between the lymphatics and subcutaneous veins at 12 months after surgery in >70% of the patients with peripheral lymphedema [96]. The reported complication profile for this procedure showed a 3.9% incidence of infection, 4.1% incidence of lymphorrhea, and 10% incidence of requiring a subsequent operation [97].

In more advanced cases of lymphedema, all main lymphatics are obstructed and tissue fluid accumulates in the interstitial spaces, spontaneously forming “lakes” and blind, sometimes interconnected “channels” of irregular shape [98]. In order to create artificial pathways for edema fluid to flow away to nonobstructed regions, Olszewski et al. suggested forming such pathways by the subcutaneous implantation of silicone tubes from the lower leg to the lumbar or hypogastric region and confirmed a rapid decrease in the calf circumference from the day of implantation over the subsequent weeks by a mean 3% with stabilization [92].

Another microsurgical procedures involving transplantation of lymphatic tissue is VLNT [98]. This surgery involves the microsurgical transfer of a small number of lymph nodes and surrounding tissue from other parts of the body, called donor sites, to the area affected by lymphedema. VLNT surgery has been described in the literature as effective in reducing the swelling and other symptoms with lymphedema [99]. Although the mechanism of action has not clearly been elucidated, the theory behind VLNT is that transplanted lymph nodes reduce lymphedema by venous shunting of lymphatic fluid and stimulating lymphangiogenesis [100,101].

In a systematic review of 18 studies and 305 patients undergoing VLNT, reduction of limb volume was noted in 86% of patients with lymphedema and 60% of patients demonstrated moderate or significant improvement of flow [102]. Through careful patient selection and meticulous technique, VLNT may result in superior outcomes to LVA for the treatment of lymphedema, especially in those with lower extremity and later stage disease [103].

However, complications of VLNT include donor site morbidity, such as seroma and lymphcele formation, delayed wound closure, and donor site lymphedema [104,105]. Other complications of VLNT are infection (7.8%), lymphorrhea (14.7%), re-exploration surgery (2.7%), and revision surgery requiring additional procedures (36%) [97].

Allen et al. therefore recommend that LVA should be performed for stages 1 and 2 patients and VLNT should be performed for stage ≥2 patients with no functioning lymphatics on ICG fluorescent imaging [100].

### 5.5. Liposuction and Surgical Excision

Liposuction involves the removal of fat and fibrosis with suction technique [106]. Lipectomy addresses the solid component such as fibrosis and hypertrophied subcutaneous adipose tissue, that typically presents later as chronic, nonpitting lymphedema (in later stages 2 and 3 disease) of the extremities after the fluid component has been conservatively drained [97,107]. Contraindications include active cancer, infection, wound and insufficient conservative management [108]. If there is more than 4 mm to 5 mm of pitting edema in the affected extremity, the patients should not receive liposuction, as it is for removing fibrotic adipose tissue, not fluid [109].

During suction-assisted lipectomy, a tourniquet is applied to the affected extremity [110]. Liposuction is performed with suction cannulas through multiple incisions. After starting distal to the tourniquet, liposuction is performed circumferentially and in a longitudinal direction to the extremity in order to minimize damage to the remaining lymphatics [111]. Liposuction is continued until it is performed past the tourniquet and the maximal amount of adipose tissue has been removed.

Lamprou et al. reported that circumferential suction-assisted lipectomy (CSAL) reduced excess limb volume by an average of nearly 80% in patients with primary advanced lymphedema, and around 100% among those with secondary lymphedema [110]. Furthermore, liposuction has been proven effective in preventing lymphedema-associated cellulitis. When used complementarily with conservative treatment modalities or procedures improving lymphatic drainage (i.e., LVA or VLNT), the benefits of liposuction can be maintained long-term [32,112].

However, it is important to remember that liposuction is not a cure for lymphedema, as it does not treat the underlying cause of lymphedema, such as lymphatic stasis and obstruction [106]. Thus, following lifelong compression therapy, therapist treatment and microsurgery (LVA or VLNT) must be received in order to prevent the reaccumulation of lymphedema. LVA and VLNT are said to prevent fluid reaccumulation and reduce the need for compression garment use after liposuction has removed the solid component of the lymphedema [32].

A more radical debulking procedure surgery for severe lymphedema (elephantiasis) was reported by Charles in 1912 [113]. It involved surgical excision to treat scrotal and lower extremity lymphedema, with the skin and subcutaneous tissues excised circumferentially to the level of the deep fascia and the resultant wound repaired using a split-thickness skin graft. Several modifications of the Charles procedure have been reported [114,115]. This procedure is indicated not only for severe lymphedema but also as a last-ditch effort before amputation [114].

The major disadvantage of this procedure is that superficial skin lymphatic collaterals are removed or further obliterated [32,100]. In addition, there is significant morbidity, scarring and risk of skin graft failure with these surgeries [106,114]. Sapountzis et al. reported that the combination of the modified Charles procedure with the vascularized transfer of a lymph node flap was effective for treating of advanced-stage lymphedema [115]. Their proposed approach consisted of preserving the superficial venous system of the dorsum of the foot and the lesser saphenous vein, which were used for the venous anastomosis of the transferred lymph node flap. During a mean 14 months of follow-up, no recurrence of lymphedema was observed.

### 5.6. Prevention of Secondary Lymphedema

We previously noted that the number of lymph node removed (OR: 3.37) and CIN removal (OR: 3.92) were independent risk factors for lower extremity lymphedema [4]. Thus, the introduction of a sentinel node (SLN) biopsy for select cases of gynecologic cancers might minimize the risk of lymphedema or prevent of lymphedema [2]. Hareyama et al. compared the incidence of LLL in patients with CINs removed or preserved [116]. The incidence of LLL was shown to be significantly lower in the group with preserved nodes; furthermore, no patients in the preserved group had lower extremity cellulitis. The preservation of CIN might therefore also minimize the risk of lymphedema or prevent it altogether.

Patients who have undergone radical surgery for gynecologic cancer must practice a careful lifestyle in order to prevent LLL [117]. The incidence of lymphedema has been found to be lower in patients who exercise regularly, receive lymphedema education before treatment and perform preventive self-care activities than in others [118]. Meticulous skin hygiene and nail care are essential for reducing the risk of bacterial infection and cellulitis. In addition, maintaining a normal weight and eating a healthy diet are an important part of healthy survivorship, as the risk of lymphedema increases in those who are overweight or obese. Furthermore, gentle limb exercises (slow and methodical) should be begun as soon as possible after surgery or radiation therapy. Patients should also avoid sun and heat while travelling—never stay sitting for long periods, and wear long-sleeved cotton garments to protect their limbs from bites or burns. Moreover, patients should keep their arms or legs elevated above the level of the heart when possible and avoid constricting the affected limb [119].

## 6. Discussion

Venous capillaries reabsorb 90% of the interstitial fluid and lymphatic channels absorb the remaining 10% of lymph fluid and proteins [16]. In cases of watery leg edema, such as during pregnancy, the methods of water immersion, bandage and stockings mainly move interstitial fluid from the extravascular space into the venous system [73], and foot edema can easily be managed. In cases of lymph edema, lymph fluid and protein do not move from the extravascular space into the venous system. Therefore, to treat lymphedema, we must move lymph fluid and proteins into the extravascular space.

Once lymphedema has developed, it has been said that it cannot be cured. Thus, the treatment of lymphedema focuses on reducing the swelling and controlling the pain. In order to reduce the swelling sufficiently, lymphedema must be diagnosed and treatment started as early as possible. We therefore need tools that can help diagnose lymphedema before secondary lymphedema develops. Among the available diagnostic methods, ICG-LG can detect dermal lymph backflow sign in asymptomatic legs and the earliest splash pattern at stage 0 [70]. Thus, ICG-LG is a considered the most sensitive and useful diagnostic tool for detecting early abnormal lymph flow after the gynecologic surgery.

At symptomatic stage 1, such as during the early accumulation of fluid, a physical examination, ultrasonography, MRI, MRI-LG/CT-LG and lymphosintigraphy are also useful. Of these methods, only a physical examination may not be able to make a quantitative measurement of lymphedema. Ultrasonography, particularly portable ultrasonography, is easy to use and a very convenient tools for checking the degree of lymphedema [51]. Ultrasonography can easily detect increased skin thickness, increased subcutaneous tissue thickness, and increased subcutaneous echogenicity [52,53]. MRI may not be suitable for the diagnosis of changes in lymphedema because water-fat MRI cannot differentiate water signals between fluid and muscle [54]. Regarding MRI-LG, an intradermal injection rather than a subcutaneous injection is required for the optimal visualization of lymphatics and poor venous contamination [59]; however, this examination requires a long duration (about an hour). CT-LG enables the clear visualization of lymphatic vessels with a minimum lumen size of 0.7mm [56] within a short time (about 5 min); however, CT-LG has an undesirable radiation exposure. Lymphoscintigrams of lymphedema show a variety of abnormal lymphatic flow findings [63,64,65], although they cannot detect an abnormal early lymph flow.

To reduce the swelling and control the pain of lymphedema, we basically perform CDP, including MLD, compression therapy, exercise and skin care. These treatments are believed to be the best for managing lymphedema [72]. Most of the lymphedema volume reduction occurs in the first 10 days of CDP therapy involving MLD [74]. Because a lower PEV (early stage of lymphedema) can predict a better response to CDP therapy, treatment for lymphedema should be started at the early stage (stages 0–1) in order to reduce lymphedema completely.

Intensive CDP therapy involves MLLB [72]. Twelve hours per day is a sufficient duration to apply bandaging [78]. NMLB may also significantly reduce both swelling and swelling-related symptoms [79]. Newly developed intensive therapy regimens also involve APCDs, which offers several approaches to managing lower extremity swelling, producing a gentle, wave-like application of pressure as the chambers sequentially inflate and deflate [83]. Of note, the new channels inflate proximally, enabling the moved fluid to be absorbed in areas such as the upper thigh, hip and lumbar region [84]. The earlier such treatments are started, the better the effect on reducing edema.

LVA procedures for stages 1–2 patients have been developed in recent years. Regarding its indication, this microsurgical procedure may be considered for patients who are refractory to conservative treatment. LVA can be performed if some portions of the peripheral collecting lymphatics remain patent and partially drain tissue edema fluid [92]. Lymphatic obstruction is more likely with LVEEA than with LVSEA. Therefore, when LVA is performed, LVSEA is principally recommended [95].

At stage 2-3, VLNT is useful for lymphedema treatment [100,103]. VLNT is thought that transplanted lymph nodes reduce lymphedema by venous shunting and stimulating lymphangiogenesis [100,101]. In cases of lymphatic elephantiasis, liposuction or surgical excision is performed to reduce the excess limb volume. LVA and VLNT are said to prevent fluid reaccumulation and reduce the need for compression garment use after liposuction has removed the solid component of the lymphedema [112]. Regarding surgical excision, the combination of the modified Charles procedure with vascularized transfer of the lymph node flap is effective [115].

These ingenious approaches can reduce leg swelling and pain and maintain the QOL of afflicted patients. However, such techniques are limited as treatments because they cannot cure lymphedema completely. Thus, the most important point is preventing the occurrence of secondary lymphedema altogether. After gynecologic cancer surgery or radiation therapy, patients must diligently conduct skin care, weight control, gentle limb exercises, avoidance of the sun and heat and elevation of the affected leg in order to prevent the start of abnormal leg lymph flow [119]. Furthermore, to prevent lymphedema, the American Cancer Society also suggests that patients ‘Get regular medical check-ups,’ ‘Report any changes to your doctor,’ ‘Try to get and/or stay at a healthy weight,’ ‘Do exercise (Using your muscle also helps the lymph fluid drain like it should,)’ ‘Try to avoid infections, burns, and injuries,’ ‘Try to avoid constriction of the leg,’ ‘Be aware of cellulitis,’ and ‘wear compression garments on long or frequent air travel’ [120].

Using preventive methods, we must prevent progression to lymphedema. The above-mentioned risk-reducing behaviors should help to reduce the production of lymph, which is directly proportional to the blood flow, and minimize blockage of lymph transport means [121,122]. Sun, heat, infections and injuries increase blood flow and thus lymph production in the leg, while sitting for long periods and compression garments may result in obstruction to lymph flow [121].

The duration of the stage 0 (or 1a) period, which refers to a latent or subclinical condition, is thought to be months or years [37]. The average time from gynecologic cancer treatment to the development of symptomatic lymphedema was reported to be 4.75 years [123]. After that, the condition progresses to advanced stages over time. Patients and physician need to be aware of this potential long-term iatrogenic complication in order to avoid a delay in the diagnosis and administration of therapy.

Medical teams should be concerned with not only curing gynecologic cancer cures but also preventing and managing side effects as early as possible. We should instruct our teams on preventive methods more carefully and eagerly.

## 7. Conclusions

Lymphedema that develops after gynecologic surgery or radiation therapy cannot be completely cured and must therefore be managed for a long period of time. With the development of new diagnostic and treatment procedures, we can reduce leg swelling and pain and maintain a good QOL. For example, ICG-LG can detect an abnormal lymph flow at most early stages, making it a useful diagnostic method that enables the early start of treatment. Regarding treatments, CDP, such as bandaging and PCT, and microsurgeries, such as LVA and VLNT, have been developed. However, such techniques are limited as treatments because they cannot cure lymphedema completely. Thus, the most important point is preventing the occurrence of secondary lymphedema altogether. Medical teams should be concerned with not only curing gynecologic cancer but also preventing and managing lymphedema and instructing patients concretely on how to improve their QOL.

## Figures and Tables

**Figure 1 healthcare-07-00101-f001:**
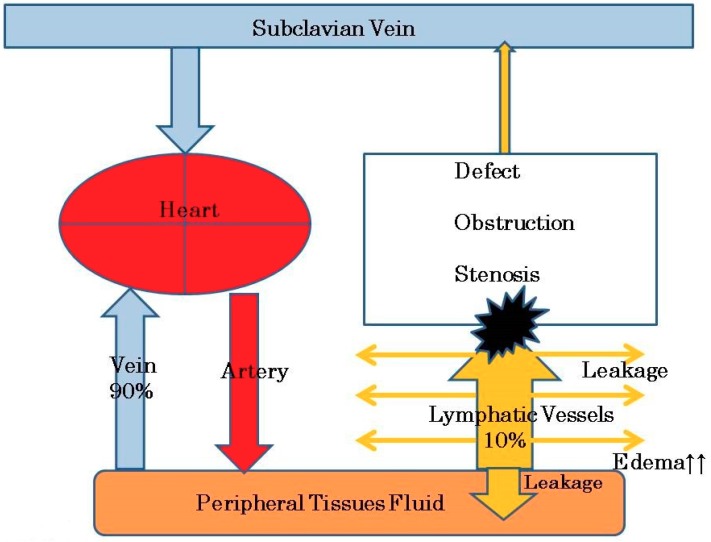
Under normal conditions, venous capillaries reabsorb 90% of the interstitial fluid and lymphatic channels absorb the remaining 10% of lymph fluid and proteins. Pelvic lymphadenectomy or irradiation can induce the destruction or obstruction of the central lymphatic vessels. Lymph stasis results in the accumulation of protein in the extracellular space, which increase the tissue colloid osmotic pressure, causing edema formation.

**Table 1 healthcare-07-00101-t001:** International Society of Lymphology (ISL) stage of lymphedema and clinical manifestation [37].

ISL Stage	Clinical Manifestation
**0 (or 1a)**	Swelling is not yet evident despite impaired lymphtransport, subtle alterations in tissue fluid/composition, and changes in subjective symptoms.
**1**	Pitting may occur. An early accumulation of fluid relatively high in protein content occurs, which subsides with limb elevation.
**2**	Limb elevation alone rarely reduces the tissue swelling and pitting is manifest.
**Later in Stage 2**	The limb may not pit as excess subcutaneous fat and fibrosis develop.
**3**	Lymphostatic elephantiasis occurs, where pitting can be absent and trophic skin changes such as acanthosis can be found.

ISL = International Society of Lymphology.

**Table 2 healthcare-07-00101-t002:** Diagnostic procedures for secondary leg lymphedema and their features.

Diagnostic Procedure	Features
Physical Examination	It may not be able to make a quantitative measurement of lymphedema.
Ultrasonography	Ultrasonography shows increased skin thickness, increased subcutaneous tissue thickness, and increased subcutaneous echogenicity. [52]
MRI	Water-fat MRI fails to differentiate water signals between fluid and muscle, which makes it difficult to evaluate subfacial tissue changes. [54]
MRI-LG	There are some limitations associated with MRI-LG, such as the long duration (about an hour) of the MR examination. [57]
CT-LG	CT-LG images can be obtained within a short time (about 5 min.). However, CT-LG has a disadvantage in its associated radiation exposure. [56]
Lymphoscintigraphy	Lymphoscintigraphy shows obstruction with visualization of discrete lymphatic trunks and slow transport. [65,66]
ICG-LG	ICG-LG detects the dermal lymph backflow sign in asymptomatic limbs and the splash pattern at the earliest stage. [68]

MR I = magnetic resonance imaging; MRI-LG = magnetic resonance lymphography; CT-LG = computed tomography lymphography; ICG-LG = indocyanine green lymphography.

**Table 3 healthcare-07-00101-t003:** LDB stage based on ICG lymphography findings. [70].

LDB Stage	ICG-LG Findings
satge 0	Linear pattern only
stage 1	Linear pattern + Splash pattern ^a^
stage 2	Linear pattern + Stardust pattern (1 region) ^b^
stage 3	Linear pattern + Stardust pattern (2 region) ^b^
stage 4	Linear pattern + Stardust pattern (3 region) ^b^
stage 5	Stardust pattern (associated with Diffuse pattern)

ICG-LG= indocyanine green lymphography; LDB=leg dermal backflow; ^a^ Splash pattern is usually seen around the groin. ^b^ Lower extremity is divided into three regions: the thigh, the lower leg and the foot.

**Table 4 healthcare-07-00101-t004:** ISL stage and treatments.

ISL Stage	Treatment
**0 (or 1a)**	None
	CDT [7,72,74]
**1**	CDT [7,72,74]
	LVA [92,94,100]
**2**	LVA [92,94,100]
	VLNT [100,103]
**Later in Stage 2**	VLNT [100,103]
	Liposuction [97,107]
**3**	VLNT [100,103,111,112]
	Liposuction [97,107], Surgical Excision [113,114]

ISL = International Society of Lymphology; CDT = complex decongestive physiotherapy; LVA = lymphaticovenous anastomosis; VLNT= vascularized lymph node transfer.

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
