# Peer review of "Development and Themes of Diagnostic and Treatment Procedures for Secondary Leg Lymphedema in Patients with Gynecologic Cancers"

_healthcare, 2019, doi:10.3390/healthcare7030101_

Round 1

Reviewer 1 Report

Review for Manuscript healthcare-576299-peer-review-v1

General Comments: Very well written review with extensive insight and references. A few small comments are listed below.

More Specific Comments:

Title – None

Abstract

Line 19 – Change “to feel leg” to “feeling” Line 24 – Remove “sign” Line 37 – Insert “the” before “affected”

Body

Line 55 – Insert “early” before “treatment” Line 95-97 – The sentence beginning with “Stated differently” needs to be reworded around the “ensure” part Line 222 – Change “palpitation” to “palpation” Line 280 – Remove “been” Line 317 – Change “circumference of calf” to “calf circumference” Line 318 – Insert “the” before “upper” In the Discussion and Conclusions section, there are text margin alignment problems.

Author Response

Review for Manuscript healthcare-576299-peer-review-v1 Answers with red color.

General Comments: Very well written review with extensive insight and references. A few small comments are listed below.

More Specific Comments: Title – None

Abstract: Line 19 – Change “to feel leg” to “feeling” Line 24 – Remove “sign” Line 37 – Insert “the” before “affected”

**Thank you for good advices. Yes, we changed “to feel leg” to “feeling”, removed “sign” and inserted “the” before “affected”.

Body: Line 55 – Insert “early” before “treatment”

**Yes, we inserted “early” before “treatment”.

Line 95-97 – The sentence beginning with “Stated differently” needs to be reworded around the “ensure” part **Thank you for good advice. we changed “Stated differently” to “In other words,”.

Line 222 – Change “palpitation” to “palpation” 

** Yes, we changed “palpitation” to “palpation” .

Line 280 – Remove “been”

**Yes we removed “been”.

Line 317 – Change “circumference of calf” to “calf circumference”

**Yes, we changed “circumference of calf” to “calf circumference”.

ILine 318 – Insert “the” before “upper”

**Yes, we inserted “the” before “upper” .

In the Discussion and Conclusions section, there are text margin alignment problems.

**Thank you for a good advice! We will consult with journal office about these problems.

Reviewer 2 Report

The review is advanced and comprehensive in its components and reads well. I appreciate that the review is categorized in to several sections and describes each component with fine details and nuances.I would appreciate a graphical representation of manuscript but overall, I support this manuscript for publication.

Author Response

Review for Manuscript healthcare-576299-peer-review-v2 Answers with red color.

Comments and Suggestions for Authors

The review is advanced and comprehensive in its components and reads well. I appreciate that the review is categorized in to several sections and describes each component with fine details and nuances.I would appreciate a graphical representation of manuscript but overall, I support this manuscript for publication.

**Thank you for a good advice! Yes, we added a Figure 1 and legend.

Reviewer 3 Report

The authors present an interesting Review on leg lymph edema after gynecologic cancer surgeries. The paper is well written and structured. I have a few concerns which should be adressed:

Title: 'Recent knowledge concerning' does not really explain what the review is about. Please specifiy.

Main text:

line 51: Why less attention than BCRL

Abbreviations: Only use it, if indicated. Otherwise, it is difficult to read. 

Line 65: keep sentences short. 

Please add a paragraph about the development of lymph edema depending on time.

Within the entire text, the authors sometimes switch between writing style for a systematic review and a book chapter. Please try to write more in the style of a review.

Please add additional information to the paragraph on primary prevention. What effect does it really have. Please add exact information from studies.

Author Response

Review for Manuscript healthcare-576299-peer-review-v3 Answers with red color.

The authors present an interesting Review on leg lymph edema after gynecologic cancer surgeries. The paper is well written and structured. I have a few concerns which should be adressed:

 Title: 'Recent knowledge concerning' does not really explain what the review is about. Please specifiy.

**Thank you for a good advice! We changed “Recent knowledge concerning” to “ Development and themes of”.

Main text: line 51: Why less attention than BCRL

**Thank you. We removed “but it has received less attention than breast cancer-related lymphedema (BCRL).”

Abbreviations: Only use it, if indicated. Otherwise, it is difficult to read. 

Line 65: keep sentences short. 

**Thank you for a good advice. We divided one sentence into two sentences. ----lower-extremity lymphedema [11]. They also revealed that adjuvant---

Please add a paragraph about the development of lymph edema depending on time.

**Thank you for a good advice. We added following sentences in Discussion: “The duration of the stage 0 (or 1a) period, which refers to a latent or subclinical condition, is thought to be months or years [37]. The average time from gynecologic cancer treatment to the development of symptomatic lymphedema was reported to be 4.75 years [125]. After that, the condition progresses to advanced stages over time. Patients and physician need to be aware of this potential long-term iatrogenic complication in order to avoid a delay in the diagnosis and administration of therapy”.

Within the entire text, the authors sometimes switch between writing style for a systematic review and a book chapter. Please try to write more in the style of a review.  **We are so sorry that it is hard for us to write more in the style of a review. Because diagnosis and treatment of lymphedema are specified in guideline. In this review, we discuss new knowledge involving gudeline.

Please add additional information to the paragraph on primary prevention. What effect does it really have. Please add exact information from studies.

**Thank you for a good advice. We added following sentences in Discussion: “Using preventive methods, we must prevent progression to lymphedema. The above-mentioned risk-reducing behaviors should help to reduce the production of lymph, which is directly proportional to the blood flow, and minimize blockage of lymph transport means [123, 124]. Sun, heat, infections and injuries increase blood flow and thus lymph production in the leg, while sitting for long periods and compression garments may result in obstruction to lymph flow [123]”.
